# Does enforcing glenohumeral joint stability matter? A new rapid muscle redundancy solver highlights the importance of non-superficial shoulder muscles

Italo Belli[1,2]*, Sagar Joshi[1,2], J. Micah Prendergast[1], Irene Beck[2], Cosimo Della Santina[1,3], Luka Peternel[1], Ajay Seth[2]

**1** Cognitive Robotics Department, Technische Universiteit Delft, Delft, Zuid Holland, The Netherlands, **2** Biomechanical Engineering Department, Technische Universiteit Delft, Delft, Zuid Holland, The Netherlands, **3** Robotics and Mechatronics Department, German Aerospace Center (DLR), Munich, Germany

* i.belli@tudelft.nl

**Data Availability Statement:** The data and code supporting this study are available at https://simtk.org/projects/thoracoscapular and https://github.

## Abstract

The complexity of the human shoulder girdle enables the large mobility of the upper extremity, but also introduces instability of the glenohumeral (GH) joint. Shoulder movements are generated by coordinating large superficial and deeper stabilizing muscles spanning numerous degrees-of-freedom. How shoulder muscles are coordinated to stabilize the movement of the GH joint remains widely unknown. Musculoskeletal simulations are powerful tools to gain insights into the actions of individual muscles and particularly of those that are difficult to measure. In this study, we analyze how enforcement of GH joint stability in a musculoskeletal model affects the estimates of individual muscle activity during shoulder movements. To estimate both muscle activity and GH stability from recorded shoulder movements, we developed a Rapid Muscle Redundancy (RMR) solver to include constraints on joint reaction forces (JRFs) from a musculoskeletal model. The RMR solver yields muscle activations and joint forces by minimizing the weighted sum of squared-activations, while matching experimental motion. We implemented three new features: first, computed muscle forces include active and passive fiber contributions; second, muscle activation rates are enforced to be physiological, and third, JRFs are efficiently formulated as linear functions of activations. Muscle activity from the RMR solver without GH stability was not different from the computed muscle control (CMC) algorithm and electromyography of superficial muscles. The efficiency of the solver enabled us to test over 3600 trials sampled within the uncertainty of the experimental movements to test the differences in muscle activity with and without GH joint stability enforced. We found that enforcing GH stability significantly increases the estimated activity of the rotator cuff muscles but not of most superficial muscles. Therefore, a comparison of shoulder model muscle activity to EMG measurements of superficial muscles alone is insufficient to validate the activity of rotator cuff muscles estimated from musculoskeletal models.

com/ComputationalBiomechanicsLab/rmr-solver/
tree/v1.1 respectively.

**Funding:** This work has received support from the
Chan Zuckerberg Initiative DAF, an advised fund of
Silicon Valley Community Foundation through
grants 2020-218896 and 2022-252796. The
funders had no role in study design, data collection
and analysis, decision to publish, or preparation of
the manuscript.

**Competing interests:** The authors have declared
that no competing interests exist.

## Introduction

Health and proper mobility of the shoulder are important in performing daily activities and maintaining independence, as the shoulder enables one to lift objects, groom and get dressed, or play a sport [1]. The mobility of the shoulder permits the largest range of motion among human joints, thanks to a mechanism composed of many bone segments, joints and muscles. This impressive mobility comes at the cost of reduced stability of the glenohumeral joint (GH), which does not rely on any bony "socket", but is instead stabilized by soft tissues including muscles that span the joint [2, 3]. To preserve glenohumeral joint integrity, a balance must be reached by soft tissue forces and the contact force generated at the glenoid [4]. A subset of shoulder muscles, collectively known as the rotator cuff muscles (infraspinatus, supraspinatus, subscapularis and teres minor) are considered glenohumeral stabilizing muscles [5], which are of particular interest, as they are a common source of shoulder injuries [6]. However accessing directly rotator cuff muscles is particularly difficult via surface electromyography (EMG), as they lay deeper below other shoulder muscles. In this context, the use musculoskeletal models of the shoulder can be a powerful tool to gain insights into rotator cuff muscle function.

For musculoskeletal models that represent the complete musculature, it is challenging to predict the muscle forces required to generate specific motions, because human (and animal) joints typically feature more actuators (i.e., muscles) than movement degrees of freedom (DoFs). As a consequence, there are infinite possible combinations of muscle forces that can generate the same joint torque and acceleration, as many muscles are mechanically redundant, leading to the so-called muscle redundancy problem. This is also true for the human shoulder, which is actuated by a combination of larger surface and deeper rotator cuff muscles that span multiple joints of the shoulder complex.

Several optimization-based approaches have been employed to solve the muscle redundancy problem in biomechanical simulations, which fall into three categories: methods integrating the dynamics of the model [7–11], methods considering the model statically [12–16], and data-driven approaches [17, 18]. However, despite these efforts, there are several open issues regarding the estimation of muscle forces from a musculoskeletal model [19]. In particular, methods that integrate the dynamics of the model (such as the popularly employed CMC [8]) permit respecting physiological constraints on the activation dynamics, but are computationally expensive. Bypassing the direct numerical integration of the system dynamics with solutions like direct collocation lightens this burden but remains prohibitively slow for real-time applications [11].

On the contrary, static methods disregarding dynamic constraints and considering each instant of the movement as independent are fast but could lead to non-physical solutions, so that bounds are employed to limit the evolution of the optimization variables between consecutive time-steps [14, 15, 20]. Nonetheless, the widely used OpenSim's implementation of the static optimization approach [16] disregards the activation dynamics, and does not account for the effects of passive fiber forces. Neglecting passive forces leads to simplifications of muscle function [21], together with poor performances in estimating antagonist muscle activity at the GH joint [22]. Finally, recent data-driven machine learning methods achieved promising results [17, 18], yet they currently disregard musculoskeletal properties altogether, retaining little direct connection with the way the human body actually functions.

Altogether, the methods described above do not allow the inclusion of constraints on the joint reaction forces (JRFs) arising during the movements. A recently-developed open-access framework could enhance them to account for stability of the joint [23], but unfortunately its formulation still ignores the passive contribution of the muscle fiber when estimating the JRFs, and computations to guarantee stability remain quite costly.

In view of these reasons, a number of previous upper-extremity simulation studies concerned with shoulder function have disregarded the GH stability issue [24–27]. Other studies, given the biomechanical importance of joint stability, have overcome these limitations and included constraints on the JRF at the glenoid when estimating individual muscle forces in the shoulder [10, 28–35], or during forward dynamics simulations of shoulder movements [36]. However, most of these works have not compared how GH stability affects the estimates of individual shoulder muscle activity [10, 28, 30, 32, 33, 35], and recent investigations reported that constraining the GH force did not influence muscle activations [34].

Changes in estimated muscle activation when GH stability is enforced have been investigated during a box-lifting task [37], and significant differences were observed only in the supraspinatus. However, this analysis relied on the OpenSim's Static Optimization algorithm [16], thus neglecting passive contribution of the muscle fibers. Moreover, the formulation to enforce GH stability was not provided, and the implementation was not made available. A similar investigation was performed to understand the effects of GH stability on the estimates of muscles of the shoulder, but only in the case of serious simulated rotator cuff tears, and implementation was not released [29]. In summary, the effects of enforcing the GH stability on estimated muscle activity when resolving the muscle redundancy problem are not fully understood. Additionally, including constraints on the JRF at the glenoid has been done ad hoc and inefficiently, solving the multibody system for its reaction forces, and none of such methods is publicly available.

Therefore, the main aim of our study is to understand the effect of GH stability on rotator cuff muscles of the human shoulder from a musculoskeletal modeling approach. To do this, we developed an efficient musculoskeletal muscle activity/force solver to analyse human movement while respecting explicit physiological constraints on JRFs: we refer to this new method as the Rapid Muscle Redundancy (RMR) solver.

## Methods

To understand the effects of GH stability on estimated muscle activity of the shoulder, we employed a musculoskeletal model of the shoulder [27] implemented in OpenSim [38, 39] with experimental data and developed the RMR solver to include constraints on the GH joint reaction force (JRF) in a computationally efficient manner. Estimated muscle activations were compared to EMG and estimates from CMC (Fig 1). Unlike other static optimization methods [12–16, 20], the RMR solver estimates muscle activations and joint forces via numerical

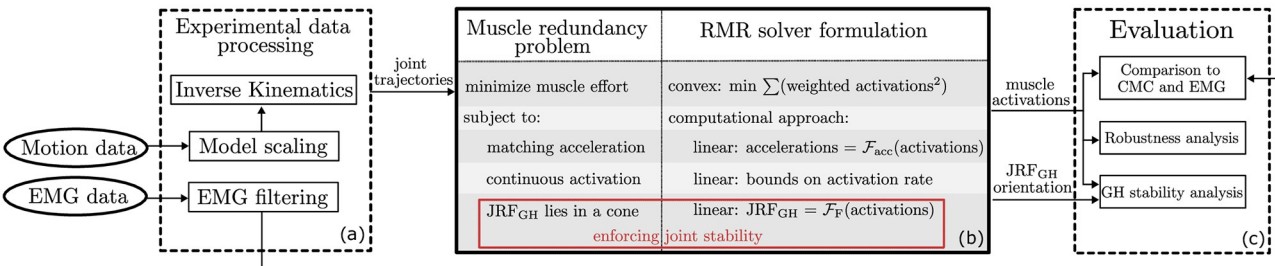

**Fig 1. Overview of the methodology to estimate and compare the effects of GH stability on shoulder muscle activity.** An inverse analysis begins with the experimental motion and measured EMG (a), which is processed via an OpenSim model to determine joint trajectories from inverse kinematics. Joint trajectories are filtered and splined to estimate joint speeds and accelerations, which serve as inputs to the Rapid Muscle Redundancy solver (b). RMR exploits an optimization problem formulation with a convex minimum effort objective, and expresses acceleration, continuous activation, and JRFs as linear functions of the activations (as design variables). We tested estimated activations against CMC and EMG, as well as inherent uncertainty, to evaluate the effect of GH stability on estimated muscle activity (c).

optimization introducing three new features: first, muscle forces include active and passive forces that are length and velocity dependent; second, consistency of activation rates are enforced using linear constraints, and third, JRFs are expressed as closed-form linear functions of the estimated muscle activation during the evaluation of the constraints. We compared our results against the state of the art and tested our conclusions for robustness against the uncertainty in motion data. We applied the RMR solver to estimate muscle activity during three shoulder tasks with and without GH stability to understand the effect of GH stability.

## Rapid muscle redundancy (RMR) solver

To tackle the main aim and facilitate the study of the effect of GH stability on rotator cuff muscles of the human shoulder, we developed a novel Rapid Muscle Redundancy (RMR) solver. The objectives for the RMR development were:

- *O1*: efficiently solve the muscle redundancy problem in a musculoskeletal model,

- *O2*: constrain the solver to realistic joint reaction forces that will maintain GH stability,

- *O3*: respect physiological constraints that include passive fiber forces, and guarantee continuity of activation profile.

The RMR solver can estimate individual muscle forces and activations given an OpenSim musculoskeletal model and movement data. The solver solves the muscle redundancy problem efficiently (*O1*) by including constraints on joint reaction forces (*O2*), passive fiber force and on the continuity of muscle activations (*O3*). The inputs to the RMR solver are the joint trajectories (angles, speeds and their accelerations) from experimental data processing (detailed below), while the outputs are estimated muscle activations and JRFs at every corresponding time point of the joint trajectories (Fig 1). Like static optimization [16], the development of the muscle redundancy problem leverages numerical optimization at each time step of the input motions. The primary difference is that more physiologically realistic behavior (*O3*) can be enforced by means of including time-dependent constraints on reaction forces and on the rate of activation change, while also accounting for passive fiber forces.

At a given time instant $t_k$, a non-linear programming (NLP) problem is formalized and solved to retrieve the optimal combination of activations $\boldsymbol{a}_k \in \mathbb{R}^{N_\mathrm{m}}$ and controls $\boldsymbol{c}_k \in \mathbb{R}^{N_\mathrm{q}}$ required to simulate the experimental motions, given the $N_\mathrm{m}$ muscles and $N_\mathrm{q}$ other actuators present in the model. We consider also the latter as, for example, it is common practice to include an ideal (reserve) actuator for each DoF to enable the model to always achieve the experimental movement in the face of modeling and/or experimental data inaccuracies, where the reserves' magnitude is an error metric [40]. The model state, including joint coordinate (angle) values and speeds $\boldsymbol{q}_k, \dot{\boldsymbol{q}}_k \in \mathbb{R}^{N_\mathrm{q}}$ for all the $N_\mathrm{q}$ coordinates, is updated once per instant with kinematics determined from the IK analysis of the experimental motion. Given the state of the model, muscle-related variables such as length and lengthening speed derived from the model can be considered fixed (for that instant), which simplifies the formalization of the constraints to the NLP problem and enables the efficient solution of the muscle redundancy problem.

**Cost function.** To find a physically plausible solution $\{\mathbf{a}_k, \mathbf{c}_k\}$ for each timestep $t_k$ of the experimental motions, we formalize our cost as a convex function to minimize the sum of weighted-squared muscle activations, to represent muscle contribution to the perceived effort:

$$J(\boldsymbol{a}_k, \boldsymbol{c}_k) = \sum_{i=1}^{N_\mathrm{m}} w_i a_{i,k}^2 + \sum_{j=1}^{N_\mathrm{q}} v_j c_{j,k}^2 \ , \tag{1}$$

where $w_i$ and $v_j$ are weightings to encourage the use of muscles over reserve actuators, and thus we set them to 1 and 10 respectively. We regularized the range of values of $c_{j,k}$ to be similar to that of the activations $a_{i,k}$, by appropriately selecting the corresponding maximum reserve forces, which are scaled by $c_{j,k}$ when producing ideal torques on the joints. In this way, the use of force generated by the ideal reserves is 10 times more costly than what is generated by the muscles. This achieves *O1* in terms of solving a muscle redundancy problem, where the simple quadratic formulation is efficient to minimize.

**Constraints.** Setting the state of the model a priori allows the formulation of simpler and more manageable constraints, which guide the solver towards a feasible solution. Muscles' paths, positions and inertia of the various bodies are also fixed based on the model's state. Recognizing that the force produced by each muscle is linear in its activation, the previous observations mean that joint accelerations depend linearly on the optimization variables as well. With some manipulation, the simulated accelerations can be constrained to match the experimental ones, by enforcing:

$$\boldsymbol{A}_{\mathrm{acc},k} \begin{bmatrix} \boldsymbol{a}_k \\ \boldsymbol{c}_k \end{bmatrix} = \tilde{\tilde{\boldsymbol{q}}}_k \ , \tag{2}$$

where $\tilde{\tilde{\boldsymbol{q}}}_k \in \mathbb{R}^{N_q}$ is obtained by subtracting the accelerations induced by gravity and passive muscle forces from the experimentally recorded accelerations, and the element $A_{\mathrm{acc},k}(j, i)$ of $\boldsymbol{A}_{\mathrm{acc},k} \in \mathbb{R}^{N_q \times (N_m + N_q)}$ represents the effect of a unitary activation/control of actuator $i$ on the acceleration of coordinate $j$. Note that this allows for correctly scaling the contribution of each actuator to the resulting coordinate's acceleration, while the effects of gravity, external forces and passive muscle forces are lumped in $\tilde{\tilde{\boldsymbol{q}}}_k$ (achieving *O3*).

Similarly, we observe that the expression for the JRF at each joint is provided by a vector sum of the moments and forces produced by each actuator. Under the same conditions reported above, we can write the JRF at a generic joint in the model as

$$\boldsymbol{F}_k = \boldsymbol{A}_{\mathrm{F},k} \begin{bmatrix} \boldsymbol{a}_k \\ \boldsymbol{c}_k \end{bmatrix} + \boldsymbol{F}_{0,k} \ , \tag{3}$$

where $\boldsymbol{F}_{0,k} \in \mathbb{R}^3$ denotes the value that the reaction force would assume at the current state if all the actuators were de-activated (accounting also for external forces, if present), and the elements of $\boldsymbol{A}_{\mathrm{F},k} \in \mathbb{R}^{3 \times (N_m + N_q)}$ account for the effect that unitary activations/controls of the individual actuators would have on the resulting components of the reaction force. Once again, the formalism adopted allows considering only the effects of the active muscle forces, while the contribution of the passive forces and gravity is lumped into $\boldsymbol{F}_{0,k}$. The stability of the GH joint is then enforced by constraining the direction of JRF at the glenoid to intersect the glenoid fossa, whose shape we approximate as circular (achieving *O2*). Similarly to [36], this results in:

$$\left( \frac{\theta_k(\boldsymbol{a}_k, \boldsymbol{c}_k)}{\theta_{\max}} \right)^2 - 1 \leq 0 \ , \tag{4}$$

where $\theta_{\max} \in \mathbb{R}$ stands for the maximum allowable angle that the reaction force can assume with respect to the line joining the glenoid center and the humeral head, and $\theta_k$ represents the angle produced by the current value of the optimization parameters. This results in a convex constraint, easier to handle in optimization. We chose a conservative value of $\theta_{\max} \approx 20°$, in agreement with cadaveric studies [4].

A final set of constraints were applied to limit muscle activation and extra actuator's controls within the bounds of physiological activation and deactivation rates. The muscle activation in the model represents the level of muscle fiber calcium release resulting from depolarization, where 0 signifies that no calcium is released and the fiber produces no active tension and 1 means maximum calcium release (activation) and that the fiber contracts maximally. Integrating the activation dynamics in [41] between consecutive time instants $t_{k-1}$ and $t_k$, we can formulate an expression that enforces activation dynamics implicitly (achieving *O3*):

$$\underbrace{a_{i,k-1} - a_{i,k-1}\left(\frac{1}{2} + \frac{3}{2}a_{k-1}\right)\frac{t_k - t_{k-1}}{\tau_{\mathrm{deact}}}}_{l_{i,k}(a_{i,k-1})} \leq a_{i,k} \leq \underbrace{a_{i,k-i} + \frac{1 - a_{i,k-1}}{\frac{1}{2} + \frac{3}{2}a_{k-1}}\frac{t_k - t_{k-1}}{\tau_{\mathrm{act}}}}_{u_{i,k}(a_{i,k-1})} , \tag{5}$$

written for muscle $i$, where $\tau_{\mathrm{act}}$ and $\tau_{\mathrm{deact}}$ are the activation and deactivation time constants of the muscle, that we set to 10 ms and 40 ms respectivey [41]. The formulation in (5) bounds the admissible activation's changes by considering the effect of minimum or maximum neural excitation input on the current activation level. The resulting $u_{i,k}(a_{i,k-1})$ and $l_{i,k}(a_{i,k-1})$ should be clipped if necessary, to always respect $a_{i,k} \in [0, 1]$.

## Shoulder musculoskeletal model

We employed the thoracoscapular shoulder model [27], which was already scaled to the subject whose movements we will consider [27]. The model (Fig 2) was previously used to capture the work done by the primary muscles of the shoulder and includes the kinematics of a 4-DoF

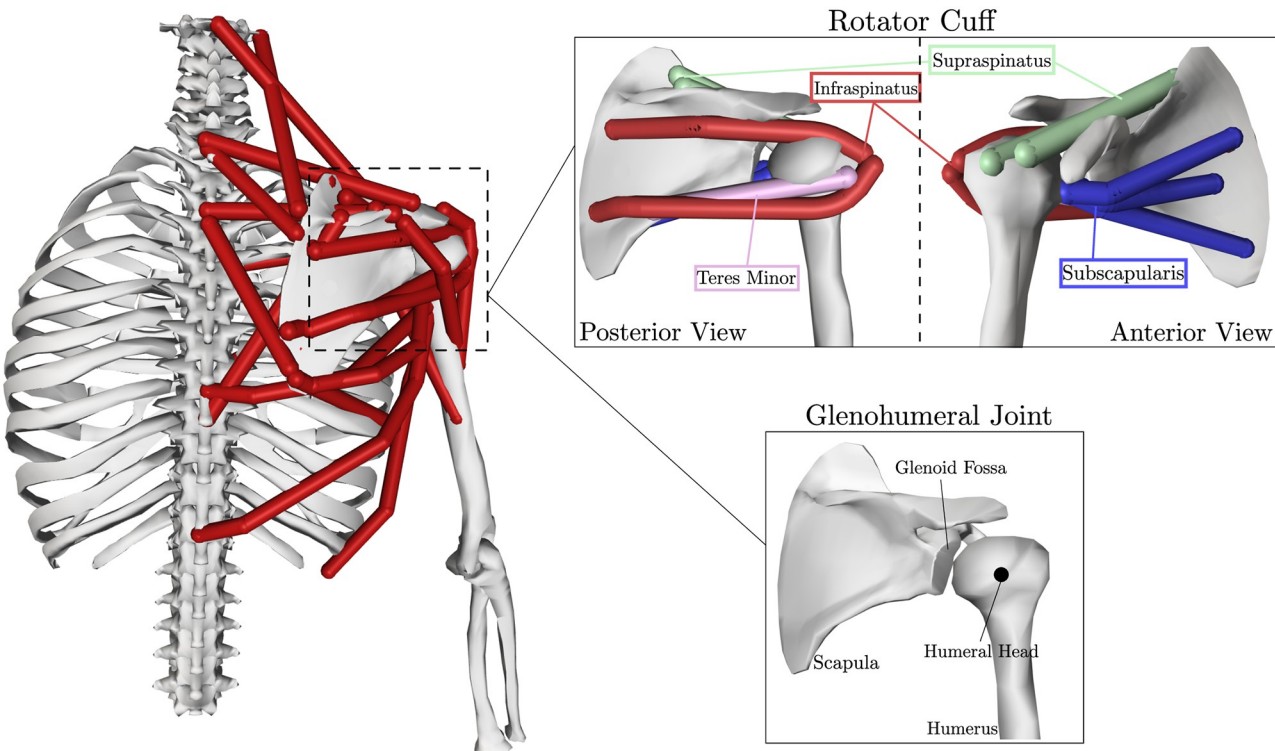

**Fig 2. The thoracoscapular shoulder model, with emphasis on the muscles (red lines), the rotator cuff, and structures of the GH joint.** In particular, the rotator cuff muscles are highlighted (infraspinatus—red, supraspinatus—green, teres minor—pink, subscapularis—blue).

scapula relative to the thorax, and the glenohumeral joint as a 3-DoF gimbal joint. The full mobility of the scapula is restricted by the clavicle by the sternoclavicular and acromioclavicular joints. Overall, the shoulder model features 7 DoFs and is actuated by 33 muscle elements.

The previous study investigated the work of shoulder muscles during common shoulder movements, but did not consider the stability of the GH joint. In this study the role of the rotator cuff muscles (infraspinatus, supraspinatus, teres minor and subscapularis, highlighted in Fig 2) is of special interest since they are anatomically considered glenohumeral joint stabilizers and their activity cannot be measured via surface EMG.

To ensure that the model successfully tracks all the experimental joint trajectories, we included ideal actuators to each coordinate in the model [40]. The estimated muscle activations are realistic if the force/moments that the extra actuators deliver are negligible compared to the muscles. We refer to all the muscles and ideal actuators as "actuators".

## Experimental motions and EMG data processing

We considered marker data and surface EMG signals for 11 muscles recorded during 18 experimental trials, at 100 Hz and 1000 Hz respectively, from a previous study [27] where the data is openly accessible. The same subject executed three repetitions of shoulder forward flexion, abduction and shrugging, with and without a 2kg load in hand, resulting in 18 acquisitions divided into 6 tasks. The electrical activity of the following muscles were measured via surface EMG electrodes and processed: anterior, middle and posterior trapezius and deltoids, pectoralis major, teres major, serratus anterior, latissimus dorsi and infraspinatus. The marker data corresponds to 9 marker trajectories, representing the motion of several bony landmarks and virtual markers extracted from post-processing of the original motion.

From experimental marker trajectories, we calculated joint angles of the shoulder model using the inverse kinematics (IK) tools in OpenSim. We were concerned about reproducing a more consistent ratio between scapula upward rotation and shoulder elevation, as reported in a previous bone pin study [50] and previously experimentally observed ratios [42]. The difference was the introduction of low scapula coordinate weights ($2 \times 10^{-4}$ for the scapula upward rotation, $10^{-4}$ for the other coordinates) to slightly penalize deviations from its neutral pose. Despite the correction, root mean square error between the experimental and simulated markers' trajectories was below 1 cm on average, across the whole dataset. The resulting joint angles were filtered with a $4^{th}$ order low-pass Butterworth filter with a cut-off frequency of 3 Hz to eliminate high-frequency noise coming from the marker data (similarly to what was done in [27]), and then differentiated to find the joint velocities and accelerations corresponding to the movement of the subject.

## Robustness analysis

To determine the effects of the GH constraint on the individual muscle activations we took into account the uncertainty of the model fit to the marker data to increase the robustness of our conclusions. Our analysis addresses the issue that for any given marker error, there are infinite solutions for joint angles that can be within that error and inverse kinematics simply provides one solution, which is not the mean or the most representative of all possible solutions. Consequently, when employing inverse kinematic to estimate the joint angle trajectories corresponding to the marker data, the uncertainty in the placement of the markers or in the scaling of the model can have a significant impact on the conclusions of a biomechanical study [43]. We focus on the uncertainty from marker placement on the model, as the uncertainty of model scaling affected derived joint angles, moments and powers in a very similar way [43].

Starting from our initial model, we generated 100 additional models by perturbing the position of each marker randomly, inside a sphere of 1 cm radius centered on the initial marker position. When generating a model, we ran inverse kinematics on all the marker trajectories in the dataset, and retained the model only if the associated average root-mean-squared error was below 1 cm [43]. We estimated the individual muscle activations of each model and associated inverse kinematics results using the RMR solver and then evaluated if muscle activations with and without the GH constraint enforced were statistically different across 3636 solutions. The 18 experimental trials for both conditions (36 solutions) were perturbed 100 times to ensure our results were robust to uncertainty from IK, for a total of 3636 solutions. To analyse the differences induced by GH stability, we employed the statistical parametric mapping (SPM) method to perform a paired t-test among the sets of one-dimensional time series describing the activations of each muscle, estimated under the two stability conditions. For every muscle, we tested the null hypothesis that the mean activation trajectory in the two conditions is the same, setting the level of significance for the test to be $\alpha = 0.01$, and leveraging the MATLAB interface of the freely available package SPM1D [44] for this analysis. Even if the two conditions achieved a p-value $<\alpha = 0.01$, this may not signify meaningful differences in muscle recruitment, and accordingly we also considered the effect size for each muscle [45]. The effect size was evaluated as the difference between the means of the activation trajectories between the two conditions. An effect size whose absolute peak value exceeded 0.1 was considered significant since such variation could corrupt significantly its match to experimental EMG values. As such, we identified a muscle to be significantly affected by the GH constraint if the requirements on p-value and effect size are both simultaneously satisfied when comparing its activations under the two stability conditions.

## Implementation and simulations

A total of 3636 RMR simulations and 36 CMC simulations were run on a Dell Latitude 7420 laptop with i7-1185G7 processor. The NLP problem (1)–(5) addressed by the RMR solver was coded leveraging OpenSim 4.3 in MATLAB R2021b and fed to the numerical solver SQP (available through *fmincon*). To cope with the gimbal lock arising in the model when the humerus is vertical, simulations of shrugging tasks were achieved locking the two indeterminate coordinates (axial rotation and plane of elevation of the humerus), which varied little through the shrugging movement and followed from previous simulations in [27]. The data used in this study is available at https://simtk.org/projects/thoracoscapular, while the code can be accessed at https://github.com/ComputationalBiomechanicsLab/rmr-solver/tree/v1.1.

## Results

We present our results beginning with the accuracy of muscle activations estimated by the RMR solver compared to experimental EMG and to CMC results as a benchmark. Next, we quantify the effect of enforcing GH stability on estimated muscle activations, while taking into account the uncertainty in the movement data.

## Comparison between RMR and CMC

Muscle activations estimated by the RMR solver and the CMC algorithm were compared to the filtered EMG signals recorded experimentally. Averaging the activation estimates over the 3 repetitions of each task, we performed a task-wise comparison. Results are reported for the RMR solver with and without the GH constraint enforced since the latter offers a more fair comparison of our solver with respect to CMC, which does not take GH stability into consideration. In Fig 3 we show, for a selection of muscles, the mean and standard deviations of muscle activations

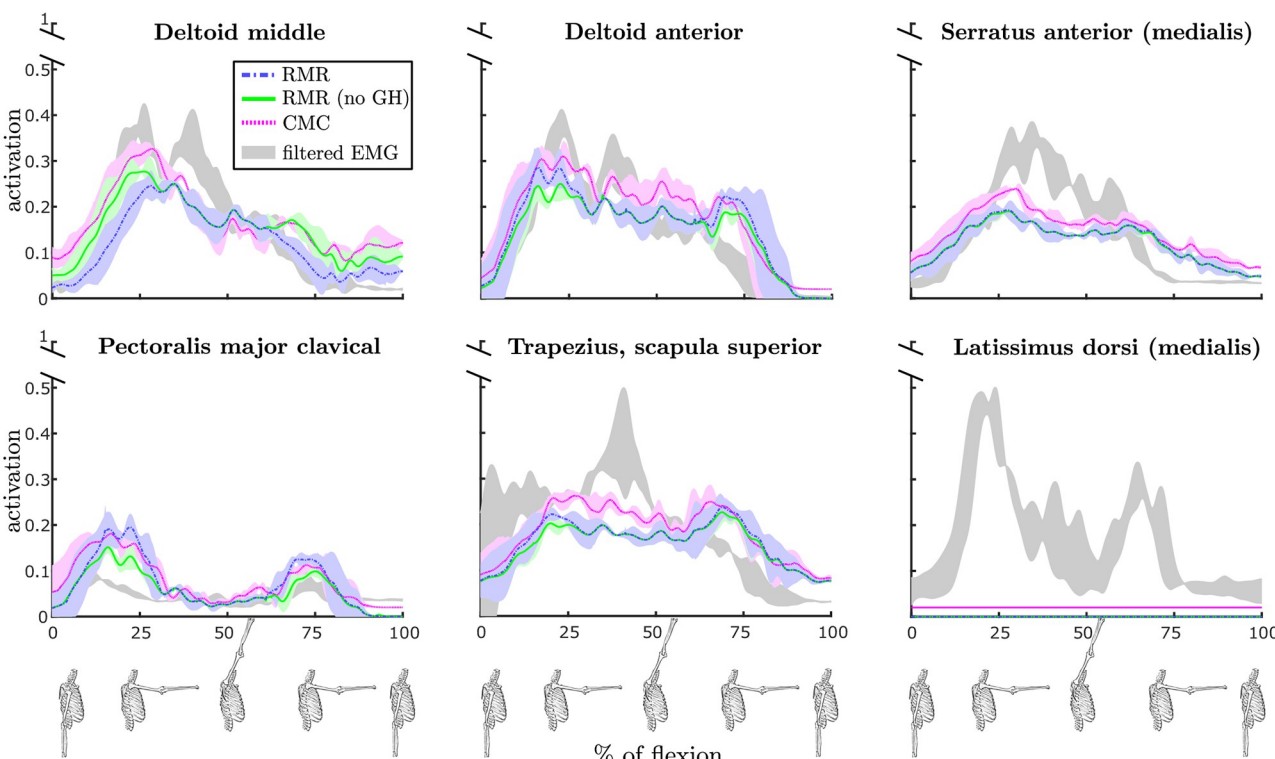

**Fig 3. Comparison of muscle activations during a *loaded flexion* movement of the shoulder as estimated by the RMR solver with (blue) and without (green) GH constraint and by the CMC algorithm (magenta).** Additionally, actual muscle activations obtained by filtered measured EMG signals are displayed in gray. For each shown muscle, estimates are displayed on vertical axes in terms of the mean across the 3 repetitions of the task (with bold lines), together with shaded ±1SD, while only the ±1SD region is reported for EMG. The horizontal axes show the progression of the movement (where 0 is the beginning and 100 is the final sample), also visually indicated with skeletal models at the bottom of the figure. The figure provides an intuitive example including the primary muscles to elevate the humerus, while an overview of all the muscles and movements is provided in Table 1.

estimated by the three methods over the duration of the task and specifically for the loaded flexion task. To capture the differences across the whole dataset, we present the mean absolute errors (MAEs) between model-estimated and EMG-recorded muscle activations over the duration of each task, and summarize them in Table 1. The values of MAEs show that activations estimated with our method and the popularly employed CMC algorithm are similar, but when visualizing the direction of the JRF at the glenoid computed by the RMR solver major changes that are induced when including GH stability in the analysis are displayed, as presented in Fig 4.

We compared the computation-to-real-time ratio for the algorithms, highlighting significant differences in performance over the whole dataset. The ratio is over 200 for the CMC algorithm, while an average close to 20 was observed for the RMR solver (both in the case in which GH stability was excluded and when it was included), making it a much more efficient option to achieve the estimates. We visualize this comparison in Fig 5, including the performances of the RMR solver when the JRF is retrieved through a force-balancing procedure by employing the OpenSim API directly, similar to [37].

## Glenohumeral stability analysis

To investigate the effect of the GH stability constraint, we compared the activations estimated by the RMR solver from the two conditions, for all the muscles. We included both the

**Table 1. Mean Absolute Error (MAE) for RMR and CMC estimates of the muscle activations against the EMG-based activations.** Results for the RMR solver are shown for both the cases in which the GH constraint was included or not. MAEs under 0.1 represent excellent matches with the experimental activations, while worse values are highlighted. For the serratus anterior, both RMR and CMC estimates are averaged over the 3 bundles composing the muscle in the model, as reported in [27].

| Task | Trapezius middle | Trapezius superior | Trapezius inferior | Deltoid anterior | Deltoid posterior | Deltoid middle | Pec.Maj. clavicle | Serratus anterior | Infra-spinatus | Latiss. dorsi | Teres major | Method |
|---|---|---|---|---|---|---|---|---|---|---|---|---|
| Abduction | 0.09 | **0.11** | 0.06 | 0.09 | 0.07 | 0.07 | 0.02 | 0.07 | 0.08 | 0.06 | 0.06 | RMR |
| | 0.09 | 0.10 | 0.06 | 0.09 | 0.07 | 0.08 | 0.02 | 0.07 | 0.05 | 0.06 | 0.03 | RMR (no GH) |
| | 0.07 | 0.10 | 0.05 | 0.08 | 0.07 | 0.07 | 0.01 | 0.06 | 0.06 | 0.04 | 0.02 | CMC |
| Abduction + | 0.07 | **0.11** | 0.05 | 0.07 | 0.10 | 0.04 | 0.03 | 0.06 | 0.05 | **0.13** | 0.08 | RMR |
| | 0.07 | **0.11** | 0.05 | 0.06 | 0.09 | 0.06 | 0.03 | 0.06 | 0.09 | **0.13** | 0.05 | RMR (no GH) |
| | 0.06 | 0.10 | 0.04 | 0.05 | 0.09 | 0.07 | 0.02 | 0.07 | 0.08 | **0.11** | 0.04 | CMC |
| Flexion | 0.06 | 0.07 | 0.04 | 0.05 | 0.05 | 0.05 | 0.02 | 0.05 | 0.04 | 0.10 | 0.07 | RMR |
| | 0.06 | 0.07 | 0.03 | 0.05 | 0.05 | 0.05 | 0.02 | 0.05 | 0.04 | 0.10 | 0.06 | RMR (no GH) |
| | 0.05 | 0.06 | 0.03 | 0.05 | 0.03 | 0.05 | 0.03 | 0.05 | 0.02 | 0.08 | 0.05 | CMC |
| Flexion + | 0.04 | 0.08 | 0.02 | 0.05 | 0.07 | 0.04 | 0.04 | 0.06 | 0.02 | **0.16** | **0.11** | RMR |
| | 0.04 | 0.08 | 0.02 | 0.04 | 0.07 | 0.05 | 0.02 | 0.06 | 0.04 | **0.16** | 0.10 | RMR (no GH) |
| | 0.03 | 0.06 | 0.02 | 0.05 | 0.06 | 0.06 | 0.04 | 0.06 | 0.02 | **0.15** | 0.09 | CMC |
| Shrugging | 0.02 | 0.09 | 0.01 | 0.01 | 0.02 | 0.02 | 0.06 | 0.01 | 0.08 | 0.03 | **0.17** | RMR |
| | 0.02 | 0.10 | 0.01 | 0.00 | 0.02 | 0.02 | 0.05 | 0.01 | 0.08 | 0.03 | 0.10 | RMR (no GH) |
| | 0.01 | 0.08 | 0.01 | 0.01 | 0.01 | 0.01 | 0.05 | 0.04 | 0.06 | 0.02 | 0.01 | CMC |
| Shrugging + | 0.04 | 0.06 | 0.02 | 0.01 | 0.02 | 0.02 | 0.05 | 0.02 | 0.02 | 0.10 | **0.12** | RMR |
| | 0.04 | 0.07 | 0.02 | 0.00 | 0.02 | 0.02 | 0.05 | 0.01 | 0.02 | 0.10 | 0.03 | RMR (no GH) |
| | 0.02 | 0.05 | 0.01 | 0.01 | 0.02 | 0.06 | 0.01 | 0.07 | 0.01 | 0.08 | 0.02 | CMC |

superficial muscles (some of which were monitored experimentally with EMG) and the deeper muscle such as the rotator cuff muscles for which there were no EMG measurements. We evaluated the set of 3636 solutions considering the uncertainty in the movement data to discern significant and meaningful differences in the activation patterns between the GH stability conditions. Fig 6 presents the mean and standard deviation of the estimated muscle activations and regions of statistical difference (shaded) and the corresponding effect sizes (below each plot) for the loaded abduction task. We are particularly interested on the effects on the rotator cuff muscles and prime movers whose activation varied the most among the two stability conditions. The muscles in which the peak effect size (the peak absolute difference between the two means) exceeds 0.1 are starred. In Table 2 we present the comparison of all the muscles in the model, summarized by the peak effect size for each task. The shaded values in the table represent the muscles whose activations are identified as physiologically different (p-value <0.01, peak effect size >0.1) between the two GH stability conditions.

## Discussion

The main aim of our study was to implement and analyze the effect of enforcing glenohumeral stability on the estimation of muscle activations during human shoulder movements. To that end, we developed a new open-source Rapid Muscle Redundancy (RMR) solver compatible with OpenSim in order to capture the GH stability constraint, include passive muscle fiber

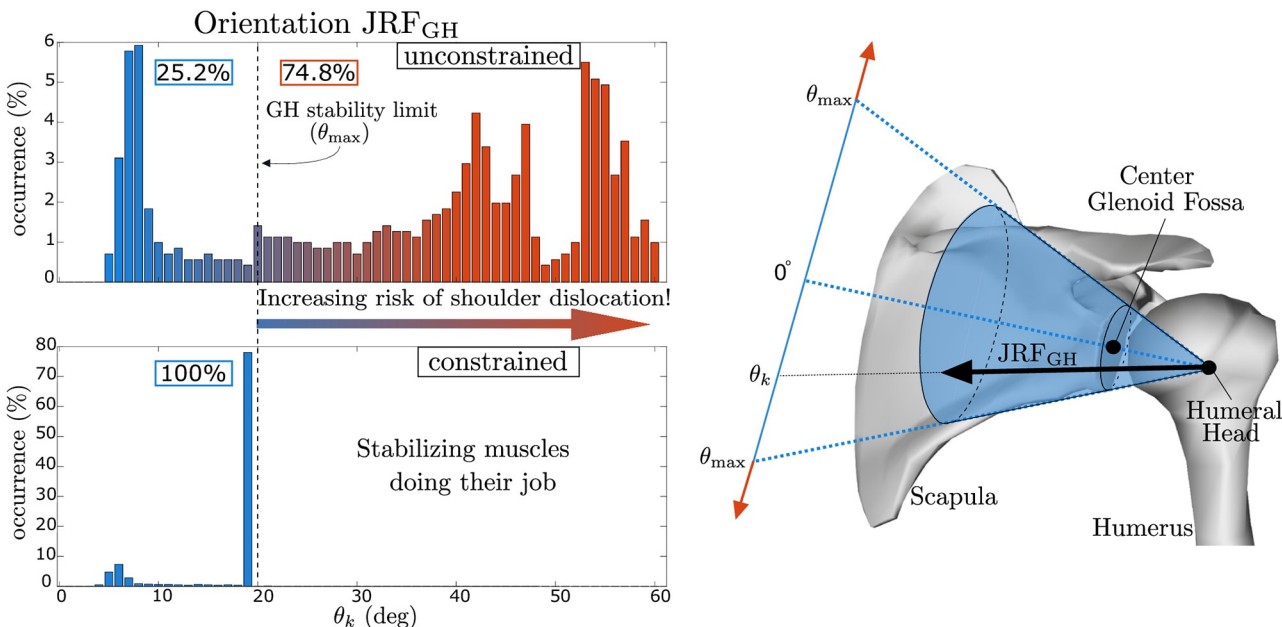

**Fig 4.** The effects of the constrained joint reaction force (JRF) at the glenoid during a loaded forward flexion (left) and the physical interpretation of the glenohumeral (GH) stability constraint (right). Since the JRF physiologically must remain within the glenoid fossa, our algorithm constrains its orientation $\theta$ accordingly. The two histograms present the values of $\theta$ in the two cases, reporting the rate (%) at which angles occur during the movement. Without ensuring GH stability, it is evident that the glenohumeral JRF can grossly exceed joint stability limits.

force contributions, and limit the physiological rate of change of estimated activations for individual muscles.

## Effect of GH stability on muscle activations

With respect to the effect of enforcing GH stability on muscle activations of the shoulder, our result showed that GH stability increased rotator cuff muscle activity (Fig 6, Table 2). Furthermore, the increase in muscle activity when accounting for GH stability is greater when the same movements are performed with a handheld weight. Interestingly, GH stability had virtually no effect on activations of surface muscles, whose EMG recordings are typically used to validate estimated muscle activity [18, 24, 27, 34, 46, 47]. Indeed our comparisons indicated

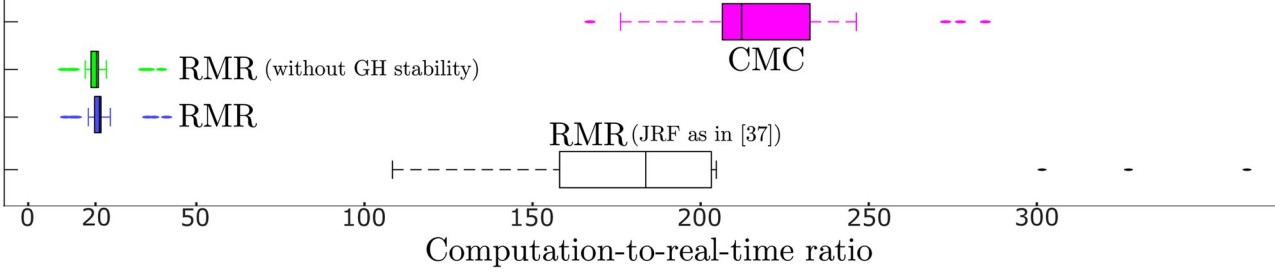

**Fig 5. Comparison of computational performance of the RMR solver versus the CMC algorithm, over the 18 experimental motions in our dataset collected at 100 Hz.** For the RMR solver, both the cases with and without the inclusion of the glenohumeral stability constraint are included and both indicate a processing rate of nearly 5 frames per second. For comparison, we include the RMR Solver formulation that includes the JRF computed from the multibody system at every constraint evaluation [37] instead of a linear function of activation at each instant.

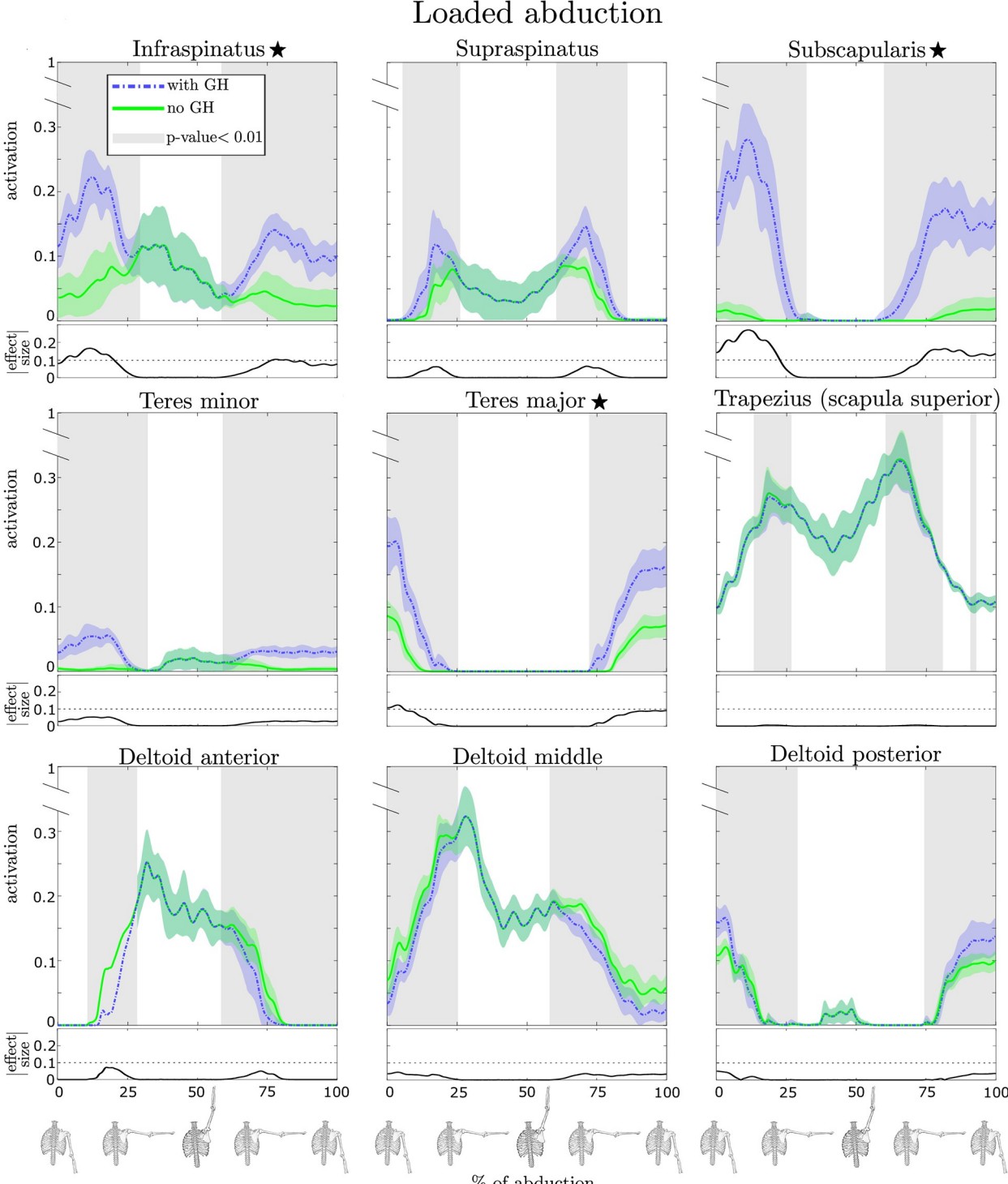

**Fig 6. Comparison of estimated muscle activations during a _loaded abduction_ obtained from the RMR solver with (blue) and without (green) GH stability enforced for several key muscles.** The vertical axes of the main graphs display muscle activation where value 1.0 is the maximum possible activation. Solid lines in the plots correspond to the activation mean while the shaded regions indicate ±1SD. The horizontal axes describe the phase of the movement where 0 is the beginning and 100 is the final sample, which are visually indicated by skeletal poses. The shaded grey sections of the phase indicate where the difference is statistically significant ($p < 0.01$). The absolute value of the effect size is shown below the main graph, and we highlight with ⋆ the muscles for which this curve exceeds 0.1. The loaded abduction task (shown) exemplified the differences between the two conditions, but an overview of all muscles for each task is provided in Table 2.

**Table 2. Peak effect size between the with and without GH stability conditions for estimated muscle activations for all shoulder muscles modeled.** Shaded values indicate that the peak effect size $\geq 0.1$ which corresponds to a meaningful difference in muscle coordination. For the multi-bundle muscles, we show the maximum peak effect size across them.

| Muscle | Flexion | Flexion+ | Abduction | Abduction+ | Shrugging | Shrugging+ |
|---|---|---|---|---|---|---|
| Infraspinatus | 0.07 | **0.14** | **0.10** | **0.17** | 0.02 | 0.06 |
| Supraspinatus | 0.04 | 0.09 | 0.03 | 0.07 | 0.04 | 0.03 |
| Subscapularis | 0.03 | **0.13** | **0.17** | **0.27** | 0.07 | 0.08 |
| Teres minor | 0.02 | 0.05 | 0.04 | 0.05 | 0.02 | 0.04 |
| Trapezius | 0.02 | 0.03 | 0.01 | 0.01 | 0.03 | 0.03 |
| Deltoid anterior | 0.03 | 0.05 | 0.02 | 0.07 | 0.01 | 0.01 |
| Deltoid middle | 0.03 | 0.08 | 0.03 | 0.05 | 0.00 | 0.00 |
| Deltoid posterior | 0.01 | 0.06 | 0.04 | 0.05 | 0.00 | 0.01 |
| Pectoralis major | 0.03 | 0.07 | 0.01 | 0.02 | 0.02 | 0.01 |
| Serratus anterior | 0.00 | 0.01 | 0.00 | 0.01 | 0.04 | 0.04 |
| Latissimus dorsi | 0.00 | 0.00 | 0.00 | 0.00 | 0.00 | 0.00 |
| Teres major | 0.04 | **0.10** | **0.12** | **0.12** | **0.13** | **0.11** |
| Levator scapulae | 0.00 | 0.00 | 0.00 | 0.00 | 0.04 | 0.02 |
| Coracobrachialis | 0.02 | 0.04 | 0.01 | 0.03 | 0.02 | 0.02 |
| Pectoralis minor | 0.00 | 0.00 | 0.00 | 0.00 | 0.00 | 0.00 |

the RMR estimates with and without GH stability were comparable to CMC with respect to filtered EMG signals, with all methods having excellent agreement (MAE $\leq$ 0.1, Table 1).

Our findings are consistent with anatomical expectations that the rotator cuff muscles act to stabilize the glenohumeral joint [2–4]. Previous shoulder modeling studies also indicated the importance of the rotator cuff muscles in controlling both magnitude and direction of the glenohumeral joint reaction force [28, 31, 35]. Yet, there are a couple of studies that observed that constraining the GH joint reaction force did not significantly affect the activations of the rotator cuff muscles [34] or observed marginal differences [37]. The models in these studies and our model, however, represent different individuals with different geometry and muscle architecture (compare Holzbaur et al. [48] to van der Helm et al. [28, 49]) and employ completely different models of scapula kinematics. We specifically model the scapulothoracic joint [50], which was attributed with improving estimates of the work done by individual shoulder muscles [27]. Differences between our studies extend to different methods to solve for the muscle redundancy, particularly in the study by Blache et al. [37], where they employed static optimization. Nonetheless, our study supports the recommendations to include a constraint on the JRF at the glenoid when investigating GH stability [34].

## Computational speed of RMR solver to estimate forces and activations

In formulating the RMR solver, the secondary aim was to improve the computational efficiency of estimating muscle forces while maintaining (or even improving) the accuracy of methods like CMC [8]. Our results indicate that the RMR solver is over an order of magnitude more efficient at estimating muscle activations in a complex musculoskeletal model of the human shoulder than the widely employed CMC algorithm.

The speed gains of the RMR solver are attributable beyond the efficiencies of static optimization. In particular, it is a result of a novel formulation of the JRF as a linear function of activation within the constraints of the RMR solver as opposed to repeatedly querying the multibody system for reaction forces in response to muscle forces [37] (Fig 5). In the case of

the shoulder, we demonstrated the efficient inclusion of JRFs by implementing the directional constraint on the glenohumeral joint reaction force.

Notably, our JRF formulation can be employed to include the reaction forces at any joint (in any OpenSim model), for example, to keep JRFs within physically acceptable bounds or to append them to the objective function to be minimized. The efficient inclusion of JRFs can be used to improve the physiological feasibility of estimated muscle forces.

The RMR solver could solve the muscle redundancy problem formulated in (1)–(5) in approximately 0.2 seconds for a single instant in time, and directly enables motion analysis at up to 5 frames-per-second. These reduced computational times pave the way for musculoskeletal model-based analysis of human motion in near real-time, which opens opportunities to apply modeling insights during rehabilitation and therapy. Our study exploited the improved computational efficiency, by enabling us to perform a robustness analysis with over 3600 trials to test against the uncertainty in the motion data. Such large-scale robustness, sensitivity and even design optimization problems would be impossible to perform in tractable time with CMC.

## RMR solver accuracy of muscle activation estimates

Estimated muscle activity from the RMR solver compared to the experimentally recorded EMG signals and, with few exceptions, the MAE observed between the two remained below the threshold of 0.1, commonly accepted as indicating excellent agreement between simulated and real muscle activation levels (see [24, 27, 51]). Reserve actuator torques were negligible for the whole duration of the movements, confirming the relative weightings selected in the cost function (1). Our results indicate that the integration of the system's dynamics performed in CMC, together with its kinematic feedback loop, may not be necessary when analyzing relatively slow movements, like the ones investigated in this study, and supports previous studies comparing static to dynamic optimization methods for estimating muscle forces in walking [52]. Our approach further supports the observation that a rigid tendon model, disregarding tendon compliance, performs just as well for a wide variety of shoulder motions [53].

## Limitations

Our results are subject to several limitations. We have modelled the GH constraint as a directional constraint, on the basis of a circular approximation of the glenoid fossa. However, this may underestimate the support provided by the actual shape of an individual's glenoid fossa and similarly by excluding soft-tissue structures such as the glenoid labrum, joint capsule, and glenohumeral ligaments in the simulations. The constraint on JRF direction in (4) results in orientations of the reaction force that lie predominantly on the boundaries of the admissible region (e.g., the perimeter of the glenoid fossa), as apparent in Fig 4. Other orientations of the JRF could be encouraged as well, but it is important to note that they would likely require higher activations than the ones we estimated for the rotator cuff muscles. Therefore, the muscle activation estimates that we report may be seen as the lower bound required for ensuring GH stability and beyond this lower bound we would expect increased influence of GH stability on the rotator cuff muscle activations. The approximation of the acceptable region based on the glenoid fossa and the subsequent directional constraint may also explain some of the outliers in our results. In particular, teres major (Tables 1 and 2) appears to be overestimated particularly during shrugging with GH stability constraints active. We believe this may be due to its role in redirecting the forces from the trapezius into the GH fossa, even though the magnitude of the force is small, which other soft tissue like the capsule and ligaments could support.

Another outlier in all cases (using CMC and RMR) was the latissimus dorsi (see Table 1), particularly during arm-raising tasks with a handheld weight, where the model underestimates muscle activity. We do not attribute these differences directly to GH stability since these discrepancies appear in both cases (with and without GH stability). Functionally, we consider latissimus dorsi as an antagonist to the deltoids and superior trapezius, and its activity when the humerus is horizontal indicates a stabilizing role in the experimental EMG data. As the model does not predict latissismus dorsi coactivation (both employing the RMR solver and the CMC algorithm, see Table 1) and instead appears to rely on particularly high activations for the teres major to stabilize the joint, we may infer that the muscle path or architecture of the latissmuss dorsi may not be captured adequately. While this is not specific to the solver, it does highlight an important point of improvement for the model, so that a proper distribution of the necessary forces could be realized.

We accounted for the uncertainty in the motion capture data, which was demonstrated to have a significant effect on the conclusions drawn in human movement studies [43]. We used the marker placement uncertainty to also account for uncertainty in model scaling. However, there is also uncertainty related to the muscle parameters and muscle architecture of the test subject, which we did not account for. Previous work concluded that estimated muscle forces and activations are also sensitive to these parameters, but personalization of the model alone does not ensure high accuracy, while properly addressing the muscle force sharing problem (as we aimed to do) is equally important [54].

## Conclusion

We showed that glenohumeral stability has a significant and large effect on the activations of the rotator cuff muscles. Our results highlight that simulation studies performed without glenohumeral stability can yield non-physical directions for the glenohumeral joint reaction force. Furthermore, the activations of the rotator cuff and teres major muscles are the most affected by the stability constraint. Consequently, we demonstrate that validation of shoulder model estimated muscle activity, by comparison to superficial EMG data, is insufficient since rotator-cuff muscle activity is unaccounted for and superficial muscles are widely unaffected by glenohumeral stability. Our results clearly show that good agreement with superficial EMG can be achieved with significantly different rotator-cuff muscle activations.

Our study has broader implications for musculoskeletal models of the shoulder and related solvers for clinical and rehabilitation applications. For example, shoulder surgery [48], robotic-led shoulder physiotherapy [55, 56], and ergonomics during manual work [57, 58] all rely on the accuracy and speed of force/activity estimates from a musculoskeletal model. Our study contributes significantly to these types of applications by providing free, open-source models and tools to accurately evaluate shoulder function in close to real-time.

## Acknowledgments

The authors thank Alexis Derumigny for his statistical-related advice, and Stephen J. Eglen from the CODECHECK project for verifying the reproducibility of our results [59].

## Author Contributions

**Conceptualization:** Italo Belli, Sagar Joshi, J. Micah Prendergast, Luka Peternel, Ajay Seth.

**Data curation:** Italo Belli.

**Formal analysis:** Italo Belli.

**Funding acquisition:** Cosimo Della Santina, Luka Peternel, Ajay Seth.

**Investigation:** Italo Belli, Irene Beck.

**Methodology:** Italo Belli, Sagar Joshi, J. Micah Prendergast, Ajay Seth.

**Software:** Italo Belli, Sagar Joshi, Irene Beck.

**Supervision:** J. Micah Prendergast, Cosimo Della Santina, Luka Peternel, Ajay Seth.

**Validation:** Italo Belli, Sagar Joshi, Irene Beck.

**Visualization:** Italo Belli, J. Micah Prendergast, Irene Beck, Luka Peternel.

**Writing – original draft:** Italo Belli, J. Micah Prendergast, Luka Peternel, Ajay Seth.

**Writing – review & editing:** Italo Belli, J. Micah Prendergast, Luka Peternel, Ajay Seth.

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
