## [Decision Letter · Decision Letter 0]

1 Sep 2023

PONE-D-23-20621Does enforcing glenohumeral joint stability matter? A new rapid muscle redundancy solver highlights the importance of non-superficial shoulder musclesPLOS ONE

Dear Dr. Belli,

Thank you for submitting your manuscript to PLOS ONE. After careful consideration, we feel that it has merit but does not fully meet PLOS ONE’s publication criteria as it currently stands. Therefore, we invite you to submit a revised version of the manuscript that addresses the points raised during the review process. The reviewers have made minor suggestions on the manuscript. Please have a look at these and clarify the methods were relevant. 

We look forward to receiving your revised manuscript.

Kind regards,

Aliah Faisal Shaheen

Academic Editor

PLOS ONE

“This work has received support from the Chan Zuckerberg Initiative DAF, an advised fund of Silicon Valley Community

Foundation through grants 2020-218896 and 2022-252796”

“The authors thank Alexis Derumigny for his statistical-related advice, and support from

the Chan Zuckerberg Initiative DAF, an advised fund of Silicon Valley Community

Foundation through grants 2020-218896 and 2022-252796.”

 “This work has received support from the Chan Zuckerberg Initiative DAF, an advised fund of Silicon Valley Community

Foundation through grants 2020-218896 and 2022-252796”

5. We note that Figure 2 in your submission contain copyrighted images. All PLOS content is published under the Creative Commons Attribution License (CC BY 4.0), which means that the manuscript, images, and Supporting Information files will be freely available online, and any third party is permitted to access, download, copy, distribute, and use these materials in any way, even commercially, with proper attribution. For more information, see our copyright guidelines: http://journals.plos.org/plosone/s/licenses-and-copyright.

Reviewers' comments:

Reviewer's Responses to Questions

**Comments to the Author**

1. Is the manuscript technically sound, and do the data support the conclusions?

Reviewer #1: Yes

Reviewer #2: Yes

2. Has the statistical analysis been performed appropriately and rigorously? 

Reviewer #1: Yes

Reviewer #2: Yes

3. Have the authors made all data underlying the findings in their manuscript fully available?

Reviewer #1: Yes

Reviewer #2: No

4. Is the manuscript presented in an intelligible fashion and written in standard English?

Reviewer #1: Yes

Reviewer #2: Yes

5. Review Comments to the Author

Reviewer #1: This is a very interesting and thoroughly executed study with clear and beneficial aims. The methods are described in sufficient detail and make reference to previous work where appropriate. Results are clearly presented and discussion is relevant and acknowledges limitations. The conclusions are justified based on the results presented. The findings of this study will be useful to researchers in the field.

Specific comments where minor improvements to clarity and detail could be made follow:

- line 43-45: Add a reference here and give an example (Are these the same papers you refer to above -ref 17,18?).

- line 74-75: re the main aim: Does it not make more sense to talk about how the RC muscle activation affects GH stability, rather than the other way around? Or perhaps you mean to study the effect of modelling choices on the prediction of RC muscle activity?

- line 86: Are you saying that these are not typically included in static optimisation methods? Because my understanding is that these are included in CMC analyses. Please clarify.

- line 133: re optimisation weights - these are somewhat arbitrary values, so are control values generally low, and can you mention this in the discussion?

- line 215-218: Not sure what this means. How were marker data adjusted? In what way is it 'updated'? Was a scapulohumeral rhythm enforced?

- line 280: "for a selection of muscles"  how were the muscles selected?

- line 411-413: Is it not more likely that the cost function does not predict co-contraction, than that the muscle path is inadequate?

Reviewer #2: I found this to be a substantial, interesting and carefully produced submission. The subject matter is increasingly relevant in a number of fields and the authors' contribution has the potential to be impactful by linking physiology, biomechanical modelling and biomechanics experiments.

Due to the quality of the submission, I will only note a number of minor issues for addressing. These follow in approximately chronological order.

The minor issues that could be addressed are as follows:

It is noted that joint angles were differentiated to find joint velocities and accelerations. Could the authors indicate how the joint angles were calculated? This is indeed given in [42] but the sequence and reference to ISB could be usefully provided here. For information, was any software used to calculate those angles (and angular velocities)? Perhaps not, but if a tool like Visual3D or OpenSim provided the calculations, it would be good to know.

3Hz is low cut-off frequency for biomechanics experiments - even in the case of smooth, slow movements. Could the authors comment [briefly] on how this is justified. Was it required for the position data or did it help to control the acceleration output, for example?

It would be useful if the authors could provide a breakdown of the 3636 solutions. Were there repetitions or is this a consequence of the number perturbations in the robustness study. Later 3600 solutions are referred to. Is this an approximation or based on 3600 solutions +36 CMC simulations.

A final, minor, observation would be that the anatomical representations of the movements at the bottom of Figures 3 and 6 are a little small. Perhaps they could be enlarged and included as separate figures of subfigures.

Overall, I thought this was a thoughtful and interesting submission.

6. PLOS authors have the option to publish the peer review history of their article (what does this mean?). If published, this will include your full peer review and any attached files.

Reviewer #1: No

Reviewer #2: No

---

## [Author Response · Author response to Decision Letter 0]

19 Oct 2023

Dear Reviewers,

we would like to thank you for raising your very interesting comments, that we have addressed specifically in the "Response to Reviewers". We appreciate your help in increasing the quality of our work, and hope that our revision will meet your expectations. 

Sincerely,

the Authors

---

## [Decision Letter · Decision Letter 1]

14 Nov 2023

Does enforcing glenohumeral joint stability matter? A new rapid muscle redundancy solver highlights the importance of non-superficial shoulder muscles

PONE-D-23-20621R1

Dear Dr. Belli,

We’re pleased to inform you that your manuscript has been judged scientifically suitable for publication and will be formally accepted for publication once it meets all outstanding technical requirements.

Kind regards,

Aliah Faisal Shaheen

Academic Editor

PLOS ONE

Additional Editor Comments (optional):

Reviewers' comments:

Reviewer's Responses to Questions

**Comments to the Author**

1. If the authors have adequately addressed your comments raised in a previous round of review and you feel that this manuscript is now acceptable for publication, you may indicate that here to bypass the “Comments to the Author” section, enter your conflict of interest statement in the “Confidential to Editor” section, and submit your "Accept" recommendation.

Reviewer #1: All comments have been addressed

Reviewer #2: All comments have been addressed

2. Is the manuscript technically sound, and do the data support the conclusions?

Reviewer #1: Yes

Reviewer #2: Yes

3. Has the statistical analysis been performed appropriately and rigorously? 

Reviewer #1: Yes

Reviewer #2: Yes

4. Have the authors made all data underlying the findings in their manuscript fully available?

Reviewer #1: Yes

Reviewer #2: Yes

5. Is the manuscript presented in an intelligible fashion and written in standard English?

Reviewer #1: Yes

Reviewer #2: Yes

6. Review Comments to the Author

Reviewer #1: I would like to thank the authors for addressing all my comments clearly and concisely. I am happy to confirm that all my concerns have been resolved.

Reviewer #2: Thank you for a very thorough response to the first review round. I have nothing further that I wish to be addressed.

Please accept my apologies for indicating that data was not available in the first version. I realised my mistake but was unable to change my review having already submitted it.

7. PLOS authors have the option to publish the peer review history of their article (what does this mean?). If published, this will include your full peer review and any attached files.

Reviewer #1: No

Reviewer #2: No

---

## [Editor Report · Acceptance letter]

21 Nov 2023

PONE-D-23-20621R1 

Does enforcing glenohumeral joint stability matter? A new rapid muscle redundancy solver highlights the importance of non-superficial shoulder muscles 

Dear Dr. Belli:

I'm pleased to inform you that your manuscript has been deemed suitable for publication in PLOS ONE. Congratulations! Your manuscript is now with our production department. 

Kind regards, 

on behalf of

Dr. Aliah Faisal Shaheen 

Academic Editor

PLOS ONE